# Antibacterial, Antioxidation, UV-Blocking, and Biodegradable Soy Protein Isolate Food Packaging Film with Mangosteen Peel Extract and ZnO Nanoparticles

**DOI:** 10.3390/nano11123337

**Published:** 2021-12-08

**Authors:** Xi Huang, Xin Zhou, Qingyin Dai, Zhiyong Qin

**Affiliations:** 1School of Resources, Environment, and Materials, Guangxi University, Nanning 530000, China; 2015391083@st.gxu.edu.cn (X.H.); 13768371312@163.com (X.Z.); wdmzsdqf@gmail.com (Q.D.); 2MOE Key Laboratory of New Processing Technology for Non-Ferrous Metals and Materials, Nanning 530000, China

**Keywords:** food packaging film, soy protein isolate, mangosteen peel extract, ZnO nanoparticles

## Abstract

The objective of this study was to prepare a functional biodegradable soy protein isolate (SPI) food packaging film by introducing a natural antimicrobial agent, mangosteen peel extract (MPE, 10 wt% based on SPI), and different concentrations of functional modifiers, ZnO NPs, into the natural polymer SPI by solution casting method. The physical, antioxidant, antibacterial properties and chemical structures were also investigated. The composite film with 5% ZnO NPs had the maximum tensile strength of 8.84 MPa and the lowest water vapor transmission rate of 9.23 g mm/m^2^ h Pa. The composite film also exhibited excellent UV-blocking, antioxidant, and antibacterial properties against *Escherichia coli* and *Staphylococcus aureus*. The TGA results showed that the introduction of MPE and ZnO NPs improved the thermal stability of SPI films. The microstructure of the films was analyzed by SEM to determine the smooth surface of the composite films. ATR-FTIR and XPS analyses demonstrated the strong hydrogen bonding of SPI, MPE, and ZnO NPs in the films. The presence of ZnO NPs in the composite films was also proved by EDX and XRD. These results suggest that SPI/MPE/ZnO composite film is promising for food-active packaging to extend the shelf life of food products.

## 1. Introduction

Owning to the low cost and easy processing plastic materials are widely used for food packaging fields [1,2]. However, the discarded non-degradable plastics have caused serious environmental pollution and endangered human and ecological safety [3]. To alleviate the threat of “plastic pollution”, a biodegradable natural polymer film is prepared to reduce or eliminate some traditional polymer packaging materials [4]. Microbial contamination is widely recognized as a key risk that not only deteriorates food but also affects public health [5]. Therefore, antimicrobial packaging materials have been hailed as one of the most promising active packaging technologies for increasing the shelf life of food goods by preventing spoiling and pathogenic microbe growth [6].

Soy protein isolate (SPI) is a plant protein that is frequently used in the food industry due to its functional properties and nutritional worth [7]. Because of the low cost, abundance, sustainability, and utility of SPI, the development of environmentally friendly protein materials with strong biocompatibility and biodegradability has piqued researchers’ interest [8]. However, SPI has low water resistance and strength, and has a lower production cost compared with synthetic plastics, causing limited application in food packaging material [9]. To enhance the strength and functionality of SPI films, blending with nanoparticles or functional additives has been proven to be an effective method [10]. ZnO NPs are generally considered safe and have good thermal stability, UV barrier, and antibacterial properties. It is noteworthy that it has excellent antibacterial activity against a wide range of bacteria, such as *E. coli* and *S. aureus*, but the effect on human cells is negligible. These advantages compared to some other nanoparticles make ZnO NPs a widely used additive in the preparation of antibacterial nanocomposite films for improving their antimicrobial properties or mechanical properties [11]. Liu et al. found that GM-based food packaging films prepared by adding the functional modifier ZnO NPs to galactomannan (GM) had good antibacterial and UV resistance properties, and their tensile strength was superior to plastic HDPE film and their barrier properties were better than those of PVC plastic film [12]. However, ZnO NPs tend to accumulate [13], thus reducing or even eliminating bacteriostatic effects. To maintain the excellent antibacterial effect of ZnO NPs, it was mixed with mangosteen peel extract to disperse evenly in the nanocomposite film. Mangosteen fruit (*Garcinia mangostana* L.), a native tropical plant, is believed to have medicinal properties. [14,15]. Mangosteen peel is high in xanthones [16], a type of polyphenolic substance with a wide range of biological activity in vitro, as well as other bioactive components such as flavonoids [17], tannins [18], and anthocyanins [19,20]. Xanthone has been demonstrated to have antioxidant [21], anti-inflammatory [22], antiallergy, antibacterial [23], anticancer [24], and antifungal properties in multiple investigations [25]. Many reports have investigated the encapsulation of various plant extracts in antimicrobial biocomposite films and their potential applications. Liang et al. prepared active film by incorporating cortex *Phellodendron* extract (CPE) with soybean protein isolate (SPI). The results showed that the film did not inhibit *E. coli* but showed a significant inhibition circle against *S. aureus*, and the inhibition circle increased with the increase of CPE content [26]. Currently, no studies have reported the combined role of MPE and ZnO NPs in SPI film active packaging applications. The SPI/MPE/ZnO composite film prepared in this work is biodegradable because the main components in the natural antimicrobial agent MPE are degradable, the SPI matrix is also degradable, and the trace amounts of ZnO nanoparticles are safe.

In this paper, biodegradable SPI-based food packaging films with excellent antibacterial, antioxidant and UV-blocking properties were prepared by the solution casting method. The effects of MPE and ZnO NPs on the mechanical properties, water susceptibility, surface color, UV-visible light barrier, antioxidant, and antibacterial activities of SPI films were investigated. Morphology, chemical structural, and thermal studies of the films were also characterized by SEM, ATR-FTIR, XPS, XRD, and TGA. It has guiding significance for the preparation of antibacterial nano-composite food packaging film. 

## 2. Materials and Methods

### 2.1. Materials

Soy protein isolate (protein content > 90%) was obtained from Henan Yuzhou Biotechnology Co., Ltd. (Zhengzhou, China). Sodium carboxymethyl cellulose (Na-CMC) was provided by Chengdu Kelong Chemical Co., Ltd. (Qionglai, China). Mangosteen was purchased in the supermarket, and the peel was dried and crushed and stored at low temperature. MPE was obtained by our previous study [27].

The ZnO NPs with an average diameter of 30 ± 10 nm and the purity of 99% were generously provided by Shanghai Shengtai Biochemical Co., Ltd. (Shanghai, China) Glycerol (purity ≥ 99%) was provided by Zhiyuan Chemical Reagent Co., Ltd. (Tianjin, China). *Escherichia coli* ATCC 25922 (*E. coli*) and *Staphylococcus aureus* ATCC 25923 (*S. aureus*) were collected from Guangdong Huankai Microbial Technology Co., Ltd. (Guangzhou, China).

### 2.2. Preparation of Nanocomposite Films

The SPI was dissolved in deionized water and stirred continuously at 80 °C until complete dissolution. The SPI solution (5 wt%) was cooled at room temperature and set aside. Glycerol with a 30% dry weight of SPI was added as a plasticizer and used to reduce the brittleness of the films. Na-CMC solution (0.5 wt%) and MPE (10 wt%) were added to the mixed solution with continuous stirring. The ZnO NPs suspension was added to the above solution at 0, 1, 3, and 5 wt% concentrations. The mixed membrane solution was treated with ultrasound (40 W, 10 min) to obtain the uniformly dispersed solution. The film solution was poured into 9 cm diameter Petri dishes and dried at 45 °C and 45% RH for 24 h. All SPI-based films were stored at 25 °C and 57% RH for 72 h before testing. A schematic diagram of the preparation and experimental formulation of the nanocomposite film were shown in Appendix A and Figure 1.

### 2.3. Characterization of Films

#### 2.3.1. Mechanical Properties

The thickness of the samples was measured using a digital micrometer (EVERTE, awt-chy01, Hangzhou, China) with an accuracy of 1 μm. Thickness was measured randomly at 5 different positions, and the average value was used.

Mechanical properties of SPI-based films (60 mm × 10 mm) were determined by a tensile testing machine (Zhiqu, ZQ-990, Dongguan, China) A tensile testing machine (Zhiqu, ZQ-990, Dongguan, China) was used to measure tensile strength (TS) and elongation at break (EAB) of films according to ASTM D882. The strain rate was 0.8 mm/s and the gauge length was 50 mm. Each kind of composite film was measured in at least five replicates. 

#### 2.3.2. Water Solubility (WS)

The determination of film solubility was based on the methods of previous studies [28]. Briefly, the films (40 mm × 10 mm) were dried at 103 °C for 24 h to determine the initial weight ($W_{i}$), and then the films were immersed in 50 mL distilled water with stirring for 24 h at 30 °C. After that, the remaining pieces of the films were filtered (0.22 μm) and dried at 103 °C for 24 h. Finally, the final weight of the film was measured, which was calculated according to the following Equation (1): (1)$$WS(\%)=\frac{W_{i}-W_{f}}{W_{i}}\times 100\%$$
where: $W_{i}$ = Initial weight of dry film (g), $W_{f}$ = Final weight of dry film (g).

#### 2.3.3. Water Vapour Permeability (WVP)

The WVP of the film was examined according to the method of Zhang et al. [15]. The weighing bottle and anhydrous calcium chloride (CaCl_2_) were placed at 105 °C for 24 h. 10 g CaCl_2_ was poured into the weighing bottle, then sealed with the film and stored at 25 °C, and 90% RH. The bottles were weighed each 1 h for 12 h. The WVP (g mm/m^2^ h Pa) of the film was calculated as follows (2):(2)$$WVP=\frac{\Delta W\times d}{t\times A\times \Delta P}$$
where: ΔW (g) is the weight increment of the bottle, d (mm) is the film thickness, t (h) is the time interval, A (m^2^) is the effective area of film, ΔP (Pa) is the part of water vapor pressure difference across the film. Three replicates were set for each sample.

#### 2.3.4. Water Contact Angle (WCA)

The hydrophobicity of the surface of the film was measured with a water contact angle tester (Fangrui, JCY-1, Shanghai, China) at 25 room temperature and 57 ± 2% RH. Sample films (4.5 cm × 1 cm) were placed on a horizontal black Teflon-coated steel stage on the analyzer. The distilled water (5 μL) was dropped on the surface of the film with a digital microsyringe, and the WCA was measured immediately.

#### 2.3.5. Films Transmittance and Opacity

The transmittance spectra and opacity of the SPI films were measured by Shimadzu UV-2450 UV-vis spectrophotometer at scanning wavelength in the range of 200–800 nm. The films (4.5 cm × 1 cm) were first dried in a desiccator for 48 h and then measured three times with a UV-vis spectrophotometer. The optical property of edible films was calculated by the following Equation (3).
(3)$$Opacity(\%)=\frac{Abs600}{L}$$

Here: Abs600 = Spectrophotometric absorbance value at 600 nm wavelength. L = Thickness of the film (mm).

#### 2.3.6. Colour Measurement

The color of the film was measured with a colorimeter (ADCI-60-C, Beijing, China) and the white plate was used as a control, and the L*, a*, b*, and ΔE values were recorded. Each film was measured at least 3 times.

#### 2.3.7. Antioxidant Activity

The antioxidant activity of the films was evaluated by measuring the free radical scavenging activity of compounds using the DPPH method. The determination of DPPH was based on our previous studies [27].

0.5 g of the film was dissolved in 10% (*v*/*v*) ethanol solution (10 mL) and shaken in a constant temperature shaker at 30 °C for 16 h. Then, the film solution was centrifuged at 6000 rpm for 10 min and the supernatant was fixed to 100 mL as the solution to be measured. 1 mL of the solution to be measured was mixed with 4 mL of 25 mg/L DPPH ethanol solution for 30 min at room temperature, and then the absorbance of the sample was measured at 517 nm.

#### 2.3.8. Antimicrobial Activity

The biocide property of the SPI films was evaluated by employing the Macrodilution method [29] recommended by previous studies. All film samples were stored in a biosafety cabinet and sterilized by UV irradiation overnight. 0.5 g of specimens were added to centrifuge tubes containing 10 mL of *E. coli* and *S. aureus* dilutions (1.0 × 10^6^ CFU/mL) respectively. The bacterial suspensions were incubated at 37 °C for 1 h. Then 100 μL of the suspensions were applied to MH agar plates and incubated at 37 °C for 24 h. The number of colonies was counted by colony counter. The Schematic illustration of the method is shown in Appendix A. The inhibition of bacteria growth was calculated as the following Equation (4).
(4)$$Reduction(\%)=\frac{a-b}{a}\times 100\%$$
where: a and b are the number of colonies of the control and test group, respectively.

#### 2.3.9. Film Characterization

The surface and cross-section morphology of the films were analyzed by a field emission scanning electron microscope (SEM, ZEISS, Oberkochen, Germany) operated at an acceleration voltage of 3 kV [27]. The energy-dispersive X-ray (EDX) spectroscopy and elemental mapping analysis (MAP) were recorded for all samples to investigate the distribution quality of ZnO NPs. The surface roughness (Rq and Ra) of the films was analyzed by atomic force microscopy (AFM, Shimadzu, SPM-9700, Kyoto, Japan). 

The thermal stability of the film samples was determined by thermogravimetric analysis (TGA) (NETZSCH, STA 449 F3, Selb, Germany). A sample of 5 mg was used and heated from 30 to 600 °C at the heating rate of 10 °C/min under a nitrogen gas flow rate of 20 mL/min [27].

The chemical structure of the films was characterized by an Attenuated total reflectance Fourier transform infrared (Shimadzu, IRTracer-100, Kyoto, Japan). The scanning frequencies were ranged from 4000 to 650 cm^−1^ with a spectra resolution of 4 cm^−1^. 

X-Ray Photoelectron Spectroscopy (XPS) was performed with Al Ka (1486.6 eV) monochromatic radiation from a Thermo Scientific K-alpha x-ray photoelectron spectrometer. Survey spectra were recorded with 1.0 eV step and 100 eV analyzer pass energy. The binding energy charge of C1s was corrected to 284.6 eV.

The crystal structure of different types of films at 40 kV and 40 mA was studied by X-ray diffractometry (Rigaku, D/MAX 2500V, Tokyo, Japan). The scattered radiation was detected in the angle range of 2θ = 5–40° with a scanning speed of 2°/min [27].

### 2.4. Statistical Analysis

The experimental data were subjected to ANOVA and Duncan’s triple range test using SPSS software to test for significance at the *p* ≤ 0.05 level (SPSS Inc. Chicago, IL, USA). 

## 3. Results

### 3.1. Mechanical Properties

The film thickness ranged from 0.10 and 0.19 mm mainly due to the different content of the filler added, as shown in Table 1. 

Compared with the pure SPI film, after introducing MPE and ZnO NPs into the SPI matrix, the TS of composite films was improved significantly. When containing MPE and 5 wt% ZnO NPs, the TS of the composite film is the highest, which is 8.84 MPa. This may be because ZnO NPs acted as a reinforcing filler in the SPI matrix, and CMC-Na contributes to the uniform dispersion of ZnO NPs in the matrix, thus improving the strength of the films [30]. P. Kanmani et al. also obtained similar results, the addition of ZnO NPs had a great influence on the mechanical properties of biopolymer films [9]. The mechanical properties of films depend greatly on the composition of the polymer, the intramolecular forces, the presence of microcrystals, and the microstructure of the film network [31,32]. Based on the SEM structural analysis of the films it was observed that as the concentration of ZnO NPs increased, it increased the crystalline structure of the SPI films by observing more continuous structures in the cross-section. In summary, the introduction of MPE and ZnO NPs in SPI films facilitates the crystalline order of the polymer and promotes the interaction between polymer chains, leading to enhanced mechanical properties of SPI/MPE/ZnO films [31].

### 3.2. WS, WVP, and WCA

The WS of SPI/MPE film was slightly reduced (*p* ≤ 0.05) compared to the control SPI film (Table 1). This may be due to the interaction between MPE and SPI chains that reduces the affinity for water molecules [33]. With the increase of ZnO NPs from 1 wt% to 5 wt%, the WS value decreased from 31.76% to 28.71% (*p* ≤ 0.05). The reduced solubility of the composite films may be related to the formation of strong structures and bonds between fillers and the protein matrix. In addition, the dimension ratio and crystalline areas of fillers also affect the water-resistance of other bio nanocomposite films [34].

The WVP of the film decreased slightly with the addition of MPE (*p* > 0.05), as shown in Table 1. This conclusion was in line with the findings of Y.A. Arfat et al. [35]. However, when ZnO NPs content was 3 wt% and 5 wt%, WVP decreased to 13.67 ± 8.74 × 10^−3^ g mm/m^2^ h Pa and 9.23 ± 4.70 × 10^−3^ g mm/m^2^ h Pa (*p* ≤ 0.05), respectively. This may be due to MPE and ZnO NPs together filling the gaps present in the structural protein chain, blocking the waterway transport pathways in the films [33]. The WVP of the composite film with ZnO NPs content of 1 wt% increased slightly, probably because ZnO NPs could separate the polymer chain and increase the free space for the water vapor to pass through [36].

The pure SPI film’s WCA was 57.80°, which is increased to 71.30° (*p* ≤ 0.05) with the addition of MPE (Table 1). The adding of a hydrophobic agent such as MPE would decrease the surface hydrophilicity and increase the WCA of the films [37]. The results showed that the polyphenol-protein interaction may change the surface energy of SPI film and enhance the surface hydrophobicity [38]. Analogous results were obtained for mango leaf extract incorporated chitosan antioxidant film [39]. When 1 wt% ZnO NPs was added into composite films, the WCA increased significantly to 75.08°, which may be related to the hydrophilicity of ZnO NPs (*p* ≤ 0.05) [9]. Moreover, the surface wettability increased with the addition of ZnO NPs [38].

### 3.3. UV-Vis Light Barrier Property, Colour, and Opacity 

The UV-visible barrier performance of the film is of great significance to light-sensitive food packaging [40]. The SPI film has the lowest UV-visible light barrier property due to the lack of UV-vis absorbent groups in the SPI structure (Figure 2a) [40]. The presence of UV-absorption phenolic compounds in MPE allowed SPI/MPE film to show stronger UV-visible light barrier property [41]. Metal oxides are widely used as UV blockers, and ZnO NPs exhibits excellent performance in blocking UV-B (290–320 nm) and UV-A (320–400 nm) [42]. SPI/MPE/ZnO films had the highest UV-vis light barrier property (*p* ≤ 0.05), which was due to the mutual aggregation of MPE and ZnO NPs impeding light transmission through the film [40]. As the concentration of ZnO NPs increases, the UV-visible light barrier performances of the films were also improved, up to 99.75%.

Five different kinds of film physical appearances were displayed in Figure 2c. Because of the yellow color of MPE, the introduction of MPE significantly reduced the lightness (L-value) and increased the yellowness (b-value) (Appendix A). Compared to SPI/MPE film, SPI/MPE/ZnO films showed lower brightness with a slight increase in a-values and a significant increase in b-values. In addition, the L-value and ΔE of the composite films had a slight decrease with the increase of ZnO NPs content, indicating a decrease in the brightness of the films. Such color change is the result of the combined action of MPE and ZnO NPs [40].

Opacity represents the amount of light that is not allowed to pass through the packaging material, and high transparency represents a low opacity value [43]. The opacity for the SPI control film was 1.07 as shown in Appendix A. The slight increase in opacity of the SPI/MPE film may be due to the interaction of SPI with MPE polyphenols to form a denser film, which reduces the amount of light passing through the film [44]. The light penetration was hindered by the combined effect of MPE and ZnO NPs, which increased the opacity of the SPI/MPE/ZnO composite films [43].

### 3.4. Antioxidant Activity and Antimicrobial Activity

Evaluation of the antioxidant property of films by DPPH radical scavenging ability and study of its change with time (Figure 3). The SPI control film showed about 43.31% antioxidant activity, which was similar to the results of the study by Z. Yu et al. [45]. The antioxidant activity of SPI film was attributed to the presence of flavonoid compounds [26]. The antioxidant property of the SPI/MPE film was significantly increased to 61.54 %, owing to the strong antioxidant effect of polyphenolic chemicals in MPE [36]. The FTIR spectra of MPE showed that there was a distinct absorption band at 3400 cm^−1^, corresponding to the phenolic group (Appendix A), indicating that MPE contained polyphenols and was a good antioxidant. HPLC analysis also showed that the main polyphenol components of MPE were γ-mangosteen, dexquinic acid, catechin hydrate, astilbin, and glucose-1-phosphate (Appendix A). However, when ZnO NPs was added, the antioxidant property of the films decreased with the increase of ZnO NPs content, which may be related to the adsorption of MPE on the surface of ZnO NPs, where MPE is immobilized on the surface of ZnO NPs and prevents free interactions with oxidized radicals [36]. In addition, the ability of the films to scavenge free radicals was found to decrease with time.

The antimicrobial activity of SPI films against *E. coli* and *S. aureus* are shown in Table 2 and Appendix A. As expected, the SPI film did not reveal antibacterial activity. However, the SPI film incorporated with MPE showed strong antibacterial efficacy against both *E. coli* and *S. aureus* (*p* ≤ 0.05). The antimicrobial activity of SPI/MPE film was attributed to the effect of phenolic compounds in MPE, which inhibited bacterial growth by disrupting the stability of the cell plasma membrane and the permeability of the cell membrane [40]. With the addition of ZnO NPs, the SPI/MPE/ZnO film showed the strongest antibacterial activity, which may be related to the combined effect of MPE and ZnO NPs [40]. The composite films containing 3% and 5% ZnO NPs showed the highest antibacterial activity against *E. coli* (100%) and *S. aureus* (97%) after 1 h, respectively. A possible mechanism for the bacterial inhibitory activity of ZnO NPs is the generation of highly reactive oxygen species (ROS) on zinc oxide surface [9]. The negatively charged superoxide and hydroxyl radicals can stay inside the bacteria’s outer cell wall, damaging proteins, lipids, and DNA, while hydrogen peroxide can penetrate the cell wall and cause cell death [9]. Compared with SPI films incorporated with ZnO NPs, the antibacterial activities arrived at the same level in the case of the different solid content of ZnO NPs (Appendix A).

### 3.5. SEM and AFM

SEM clearly shows the surface and cross-section of the films (Figure 4a). The SPI control film exhibited a smooth and dense network structure in both surface and cross-section. However, the addition of MPE led to the coarser cross-section structure of the film. The conversion of phenol to quinone induced substantial aggregation in the film matrix through non-disulfide bonds, thus controlling the roughness of the film [28]. SEM images of SPI/MPE/ZnO films showed that the higher the concentration of ZnO NPs, the smoother the cross-section structure of the film, and the more uniform the distribution of ZnO NPs in the film matrix [46]. The EDX analysis confirmed the presence of ZnO NPs in the composite films (Appendix A). As expected, the element of Zn appeared in SPI/MPE/ZnO film [47]. To show the uniform distribution of inorganic phases on the surface of SPI substrate, compositional maps (Appendix A) were recorded for atoms of interest [48]. It is clearly shown in Zn element MAP that there is no obvious gradient of Zn concentration between different regions, confirming a uniform distribution of the inorganic particles until it is up to 5% in films and when MPE is up to 10% [49].

The roughness of the film was analyzed using AFM and the results are given in Figure 4b and Appendix A. The surface of the SPI control film was somewhat rough and the addition of MPE slightly decreased the values (*p* ≤ 0.05) of Ra and Rq respectively from 11.7–6.8 nm and 15.8–9.7 nm, probably attributed to the biopolymer-extract interaction [50]. Whereas the addition of ZnO NPs increased these values from 11.7–19.7 nm and 15.8–27.1 nm, respectively (*p* ≤ 0.05). This was due to the insoluble nature of the nanoparticles and the formation of agglomeration during the film drying process [51].

### 3.6. TGA

TGA and DTG curves are presented in Figure 5a,b, respectively. Appendix A lists the maximum decomposition temperatures (Td max) and weight loss (Δw) of the corresponding films. The initial weight loss was detected between 67.44 and 100.42 °C with a weight loss of 9.63–14.15% due to the evaporation of water molecules. The subsequent steps of degradation were observed at 269.14–306.86 °C, which were due to the decrease of crystallinity and molecular weight in the blending process [52]. The effective crosslinking of phenolic compounds in MPE to SPI could improve the thermal stability of SPI films. The inclusion of ZnO NPs lowered the thermal stability of the SPI films marginally, according to TGA analysis. Thermal stability may be reduced as a result of changes in protein structure and the breaking of low-energy intermolecular interactions that maintain protein conformation [35]. This is consistent with ATR-FTIR analysis, confirming the change in the secondary structure of the protein as measured in the films. Additionally, the residual mass (i.e., char content) of all films varied between 22.39–29.63% at 650 °C. Compared to the SPI control film, the films with MPE and ZnO NPs showed higher residues, due to biopolymers containing non-ignitable minerals [53].

### 3.7. ATR-FTIR, XRD, and XPS

Figure 5c shows the ATR-FTIR spectra of SPI films. The broadband was observed at 3275 cm^−1^, corresponding mainly to the stretching of the free O-H groups and amine N-H of the SPI [45]. The addition of MPE enhanced the peak intensities at 3275 cm^−1^, 1636 cm^−1^ (amide I), and 1526 cm^−1^ (amide II), indicating that the hydrogen bonding interactions between polyphenols and proteins were weakened [54]. However, as shown in Appendix A, the ZnO NPs added to the films showed a slight shift towards the lower amide-A wavenumber (3275–3273 cm^−1^), possibly due to increased hydrogen bond formation [55]. The transition of amid-A region to low wavenumber indicated that the N-H group in the protein chain interacted with ZnO NPs mainly through hydrogen bonds [35]. 

The amide I (1600–1700 cm^−1^) was commonly used for protein secondary structure analysis [56] (Appendix A). To analyze the composition of the amide I band, curve fitting of its second derivative spectrum was needed [57]. The areas of the second derivative spectrum correspond to different types of secondary structural components [57]. In general, the bands of about 1640–1600 cm^−1^ and 1670–1690 cm^−1^ are ß-sheet characteristics, 1650–1640 cm^−1^ is unordered structures, 1660–1650 cm^−1^ is α-helical, and 1700–1660 cm^−1^ is ß-turn [54]. Table 3 shows the peak regions for each particular conformation associated with the secondary structure in the films. The content of the ß-sheet structure increased as MPE was added, while the content of α-helix and random structure of blend films was decreased. Thus, they improved tensile strength and was coincided with the finding that the content of ß-sheet contributes to the strength of the composite films, and the content of α-helix and ß-turn is related to the flexibility of the films, while the random coil structure weakens the mechanical properties of the composite films [56]. The addition of ZnO NPs into the films reduced the secondary structures (α-helix and ß-sheets) and the random structure was increased. While the random structure decreased as ZnO NPs increased resulting from order structure associated with the crystalline as is shown by XRD analysis. It suggested that the protein secondary structure in the film was gradually disrupted in the mixing with ZnO NPs.

XRD patterns of the SPI control film exhibited obvious diffraction peaks at approximately 2θ ≈ 9.0° and 20°, which corresponded to the typical α-helix and ß-sheet structures of the SPI secondary conformation, respectively, as shown in Figure 5d [58]. With the addition of MPE, new peaks did not appear, indicating good compatibility between SPI and MPE [58]. The XRD pattern of SPI/MPE/ZnO film showed obvious diffraction at 2θ ≈ 32°, 34.4° and 36.6°, corresponding to the standard (100), (002) and (101) crystal faces of ZnO metal, respectively [9], indicating that the ultrasonic blending had no impact on the hexagonal structure of ZnO NPs in the SPI hybrid matrix [59]. However, the peak intensity at 2θ ≈ 9.0° and 20° was dramatically lower than the SPI control film, indicating that the conformation of SPI molecules has changed (this could be shown from FTIR results) [58].

XPS was used to analyze the film surface, as shown in Figure 6a. For SPI/MPE/ZnO films, a pair of new ZnO 2p peaks were observed, indicating that ZnO NPs exist in the outermost layer [60]. However, the existence of Zn 2p was found too hard to be seen in SPI/MPE/ZnO film adding 1 wt% ZnO NPs. Table 4 showed the peak area measured by XPS. The peak areas of C 1s, O 1s, N 1s, and Zn 2p reflected the concentration of these atoms on the surface. When MPE and ZnO NPs were added, the proportion of C decreased, while the proportion of O increased. When ZnO NPs increased, the proportion of C increased, while the percentage of O decreased and Zn begins to appear in the spectrum, indicating that the surface composition is different [61].

The C1s spectra were deconvoluted identifying the chemical state of C detected on the surface of the film. The C1s peak was divided into three peaks (Figure 6), which are C1 (C-C/C-H), C2 (C-O/C-N), and C3 (C=O) functions [59]. Notably, the shape and position of the bands in the spectrum were similar. The binding energy values and the contents of each function are shown in Appendix A. The main peak C1 appeared at 284.8 eV, which was mainly the combination of hydrocarbon and C-C bond. C2 band represented C-O bond and C-N bond in the binding energy range of 286.2–286.3 eV, and corresponding ester group C3 was found at 288.0–288.1 eV, which was consistent with the literature [61]. The change of absorption peak reflects the change of C content in the films. C1 indicates the hydrophobicity of the surface; C2 reflects the degree of cross-linking, and C3 indicates the hydrophilicity [62]. The C1 contents of SPI/MPE/ZnO films suggested lower hydrophobicity and cross-linking degree than that of SPI control film. This can be attributed to hydrogen bonding and chemical cross-linking reactions between the SPI, MPE, and ZnO NPs [62]. The O 1s features in Appendix A represented the distribution of the oxygen present on the film’s surface. The peak value of SPI film at 531.7 eV was O-C=O/O=C-N oxygen characteristic [61]. With the introduction of MPE, the peak of O-C=O/O=C-N shifted to the higher binding energy. In addition, a new peak of O atom of ZnO NPs was found in SPI/MPE/ZnO films (532.1, 532.1, and 532.2 eV, respectively). In general, binding energy transfer in XPS spectra can be attributed to two different mechanisms: different electronegativity of metal ions and strong interactions between nanocrystals (electron transfer) [60]. Therefore, the shift of C 1s and O 1s binding energy proved the strong interaction between protein, MPE, and ZnO NPs in the composite films.

## 4. Conclusions

In this study, composite films based on SPI and MPE and ZnO NPs were successfully fabricated by a simple, green, and efficient solution casting method. The ternary blended composite films exhibited significant improvements in mechanical properties, water vapor permeability, water-solubility, UV-barrier, antioxidant property, and thermal stability. Due to the antibacterial properties of MPE and ZnO NPs, the composite films exhibited excellent antibacterial properties against *E. coli* and *S. aureus*. SEM showed that the surface of the SPI/MPE/ZnO composite film was relatively smooth, which proved that the MPE and ZnO NPs had good compatibility with the SPI substrate. According to this study, the amount of MPE (10 wt%) used provided excellent antibacterial activity, while 5 wt% of ZnO NPs provided optimum improvements in mechanical and water vapor barrier properties. Therefore, we conclude that the biodegradable SPI/MPE/ZnO composite film may serve as an ideal packaging material for food packaging applications.

## Figures and Tables

**Figure 1 nanomaterials-11-03337-f001:**
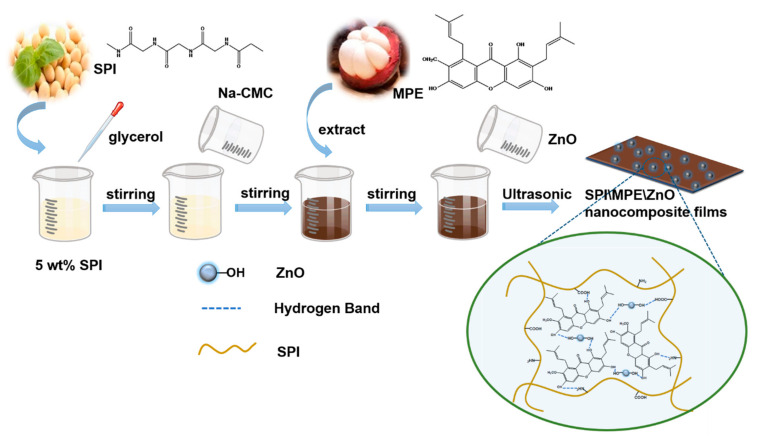
Schematic of the preparation of nanocomposite films.

**Figure 2 nanomaterials-11-03337-f002:**
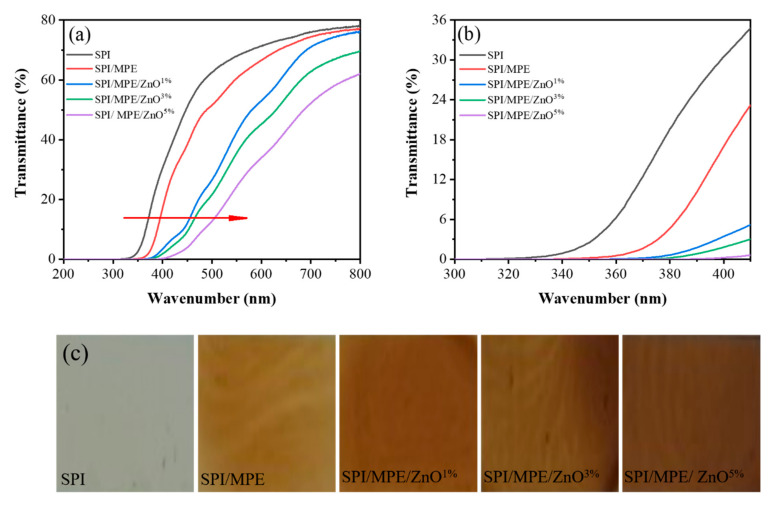
UV-vis light transmittance of SPI films. The red arrows indicate that the UV resistance of the films is significantly improved with the addition of MPE and ZnO NPs, and the increase of ZnO NPs content (**a**). UV-vis light transmittance of SPI films at 320–400 nm (**b**). The physical appearance of SPI films (**c**).

**Figure 3 nanomaterials-11-03337-f003:**
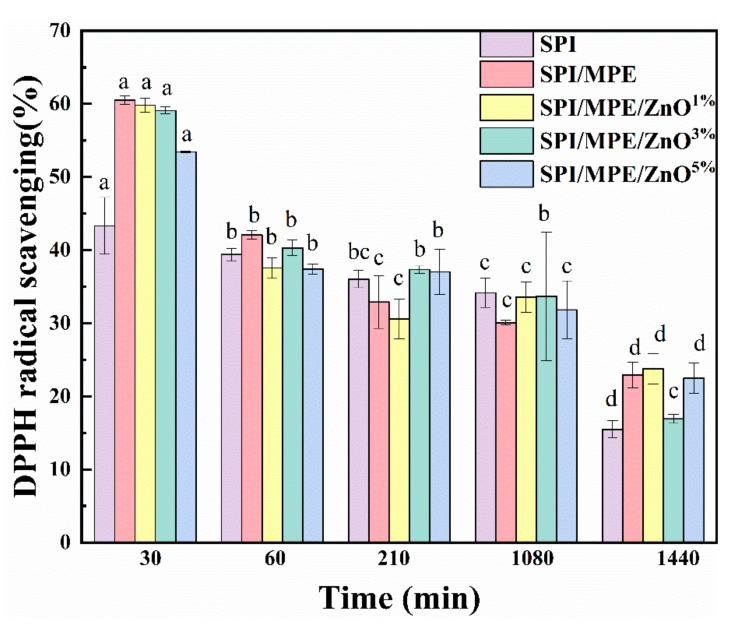
DPPH radical scavenging ability of the SPI films. All data are shown as mean ± standard deviation (SD). The superscripts different letters in a column indicate significant differences (*p* ≤ 0.05).

**Figure 4 nanomaterials-11-03337-f004:**
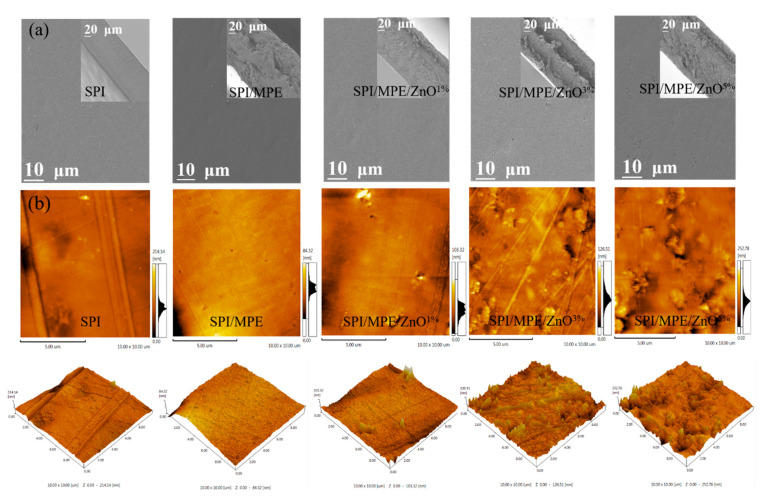
The SEM micrographs of surface and cross-section for SPI films (**a**). 2D and 3D images of surface topography for SPI films (**b**).

**Figure 5 nanomaterials-11-03337-f005:**
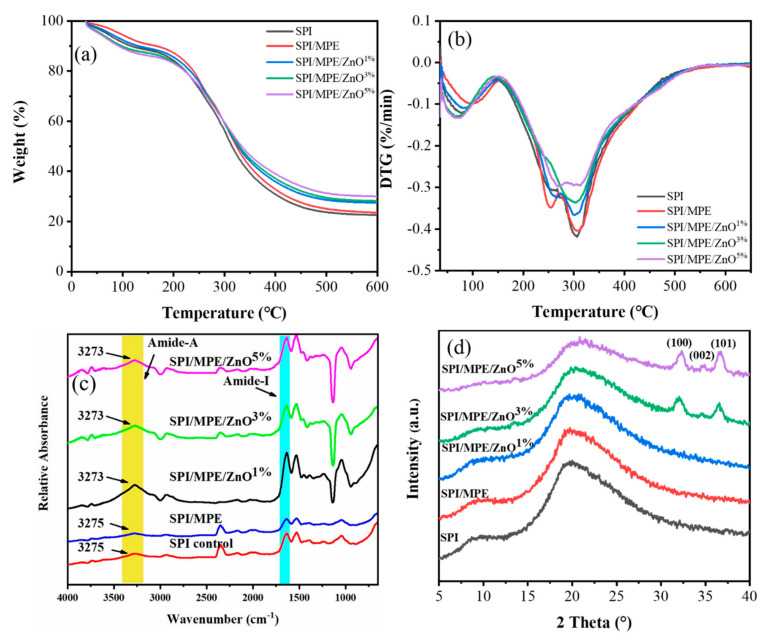
TGA (**a**) and DTG (**b**) profiles of SPI films. ATR-FTIR spectra of SPI films (**c**). XRD patterns of the SPI films (**d**).

**Figure 6 nanomaterials-11-03337-f006:**
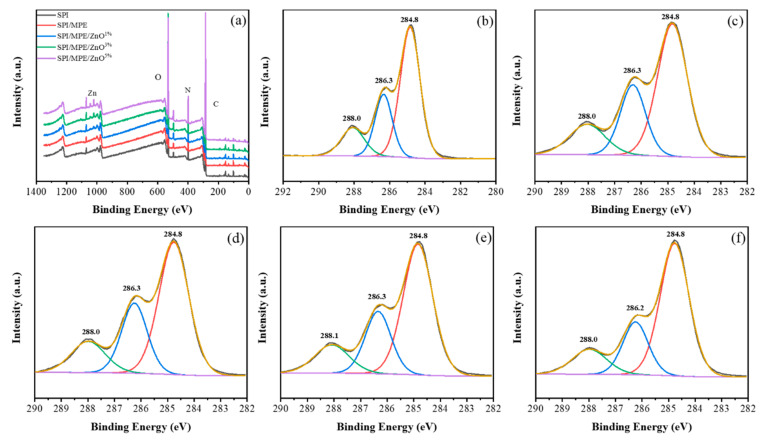
XPS survey spectra of SPI films (**a**) and XPS of C 1s features of SPI films for SPI control (**b**), SPI/MPE (**c**), SPI/MPE/ZnO^1%^ (**d**), SPI/MPE/ZnO^3%^(**e**) and SPI/MPE/ZnO^5%^ (**f**) film, respectively.

**Table 1 nanomaterials-11-03337-t001:** Physical properties and the WCA of composite films incorporated with various ZnO NPs content.

Films	Thickness (mm)	TS (MPa)	EAB (%)	WS (%)	WVP × 10^−3^ (g mm/m^2^ h Pa)	WCA (°)
SPI	0.10 ± 0.00 ^c^	5.02 ± 0.11 ^c^	82.87 ± 1.96 ^d^	41.86 ± 5.19 ^a^	19.53 ± 3.28 ^a^	57.80 ± 3.97 ^c^
SPI/MPE	0.14 ± 0.0 ^b^	6.57 ± 0.10 ^b^	59.95 ± 4.96 ^c^	33.92 ± 0.08 ^b^	19.23 ± 3.76 ^a^	71.30 ± 3.78 ^a^
SPI/MPE/ZnO^1%^	0.19 ± 0.04 ^a^	8.11 ± 0.26 ^a^	75.23 ± 27.98 ^b^	31.76 ± 0.30 ^bc^	19.67 ± 2.97 ^a^	75.08 ± 0.30 ^a^
SPI/MPE/ZnO^3%^	0.16 ± 0.0 ^b^	8.23 ± 0.31 ^a^	53.95 ± 6.68 ^b^	30.30 ± 1.69 ^bc^	13.67 ± 8.74 ^ab^	62.78 ± 0.28 ^b^
SPI/MPE/ZnO^5%^	0.12 ± 0.06 ^c^	8.84 ± 0.48 ^a^	47.88 ± 17.34 ^a^	28.71 ± 1.07 ^c^	9.23 ± 4.70 ^b^	58.75 ± 0.81 ^bc^

The values are averaged ± Standard deviation. Different letters in the same column show a significant difference (*p* ≤ 0.05).

**Table 2 nanomaterials-11-03337-t002:** Antimicrobial effect of SPI films.

Films	Bacteria	Viable Colony Numbers (CFU/mL)	Antibacterial Potency (%)
SPI	*E. coli*	926 ± 4.24 ^a^	0
	*S. aureus*	813 ± 32.53 ^a^	0
SPI/MPE	*E. coli*	730 ± 8.49 ^b^	21.17
	*S. aureus*	329 ± 6.36 ^b^	59.53
SPI/MPE/ZnO^1%^	*E. coli*	512 ± 33.23 ^c^	44.71
	*S. aureus*	57 ± 0.71 ^c^	92.99
SPI/MPE/ZnO^3%^	*E. coli*	0	100
	*S. aureus*	22 ± 1.41 ^c^	97.29
SPI/MPE/ZnO^5%^	*E. coli*	0	100
	*S. aureus*	21 ± 0.71 ^c^	97.42

The values are averaged ± Standard deviation. Different letters in the same column show a significant difference (*p* ≤ 0.05).

**Table 3 nanomaterials-11-03337-t003:** Secondary structure contents of the films.

Films	% α-Helix	% β-Sheet	% β-Turn	Random Coil (%)
SPI	13.02	51.81	18.34	13.64
SPI/MPE	12.90	52.52	21.22	13.36
SPI/MPE/ZnO^1%^	15.89	50.99	14.85	18.27
SPI/MPE/ZnO^3%^	15.13	50.59	16.99	17.29
SPI/MPE/ZnO^5%^	14.86	50.30	17.60	17.24

**Table 4 nanomaterials-11-03337-t004:** The peak area is measured by XPS.

Films	C (at. %)	O (at. %)	N (at. %)	Zn (at. %)
SPI	69.12	18.94	8.04	0
SPI/MPE	65.62	22.41	8.15	0
SPI/MPE/ZnO^1%^	65.19	22.74	8.57	0.1
SPI/MPE/ZnO^3%^	65.51	22.25	7.84	0.35
SPI/MPE/ZnO^5%^	68.14	20.67	7.43	0.52

## Data Availability

Summary data available upon request.

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
