# Peer review of "Antibacterial, Antioxidation, UV-Blocking, and Biodegradable Soy Protein Isolate Food Packaging Film with Mangosteen Peel Extract and ZnO Nanoparticles"

_nanomaterials, 2021, doi:10.3390/nano11123337_

Round 1

Reviewer 1 Report

Manuscript: nanomaterials-1456430

Antibacterial, UV-blocking and biodegradable soy protein isolate food packaging film with mangosteen peel extract and nano-zinc oxide

General comments:

This manuscript reports findings on important properties of soy protein isolate packaging films containing mangosteen peel extract and nano-zinc oxide. The produced films showed strong antioxidant and antibacterial activities with a high UV-barrier property. A number of useful characterization techniques have been used by authors to evaluate the research hypothesis. In my opinion the topic and research subject has enough novelty enabling this manuscript to be published in Nanomaterials. However, I would suggest flowing points to be considered by authors to improves the quality of this manuscript.

Title:

Please refer to antioxidant properties as well in the title.

Abstract:

Please organize your abstract according to following context: A short introduction, hypothesis (aim) of the study, methods, the most important results (quantitative), a general conclusion, and future prospective (implication of the results). Please revise English grammar thoroughly.

Introduction:

Please give a brief survey of literature about similar reports on active biocomposite films containing nanoparticles loaded antioxidant and antimicrobial agents (especially from natural origins) and explain the novelty of your research compered to those literature.

Please provide the “research hypothesis” at the end of your introduction.

You may refer to following literature in Introduction and/or Res & Disc sections.

https://doi.org/10.3390/nano11061439

https://doi.org/10.1016/j.ijbiomac.2021.10.070

Materials & Methods: 

Section 2.2.3 WVP.  Please indicate total duration of your weight recordings for WVP test.

Section 2.3.6. Colour: Please provide more details about your photography (device). Add a relevant reference.

Section 2.3.8. Revise English grammar. Also describe the test in more details.

Section 2.3.9 please provide relevant references for TGA/DSC XRD SEM analysis.

Results and Discussion:

Table 1. please reorganize. (LINES)

Caption Figure 2: please specify difference between Figs (a) and (b).

AFM results: Please provide Ra and Rq roughness parameters extracted from AFM images and indicate mean ± STDEV for these parameters among film samples.

Figure 5: TGA analysis: it seems that Figure b for DTG is not correct. DTG is a degradation temperature extracted from TGA results and gives the maximum temperature at which weight loss occurs.

Please provide DSC results based on TGA curves for better interpretation of the results.

Author Response

Dear reviewers,

    Thanks for the reviewers’ comments concerning our manuscript entitled “Antibacterial, UV-blocking and biodegradable soy protein isolate food packaging film with mangosteen peel extract and nano-zinc oxide”. Those comments are all valuable and helpful for revising and improving our paper, as well as the important guiding significance to our research. We have taken them into careful consideration and have made a correction which we hope meets with approval. Revised portions are marked in red in the manuscript. Major revisions to the paper are highlighted in blue below and responses to reviewers' comments are provided below.

    We are very grateful to the reviewers for their enthusiastic work and hope the correction will meet with approval.

    Thanks again for your comments and suggestions, which make our manuscript more rigorous and scientific.

Reviewer 2 Report

The authors need to add some more references of the applications of ZnO in the manuscript. 
The figure 5(c) may not be correct or the labeling is not appropriate. 

The authors need to employ the drug resistant strains in their antimicrobial strains.

Author Response

(The authors gave the same response as above.)

Reviewer 3 Report

This study about the incorporation of MPE and ZnO NPs in casted SPI films aimed to obtain anti-bacterial and UV-blocking properties is, very well conducted. All different aspects are investigated using the required test methods and explained in sufficient detail.

Biodegradability as claimed in the title is not investigated in the paper. As ZnO NPs (and MPE?) might also affect the micro-organisms in compost, I am not sure if it can be formulated like this in the title and conclusion without proof.

In my opinion, the study is very well performed and written, therefore I recommend publication after minor revisisons. Please find attached my comments and suggestions.

Author Response

(The authors gave the same response as above.)

Round 2

Reviewer 2 Report

The authors state "Figure 5c shows the ATR-FTIR spectra of SPI films." but there is no 5c (labeled).

Author Response

Dear reviewers,

    Thanks for the reviewers’ comment, Figure 5 has been corrected.